# Vitrification Using Soy Lecithin and Sucrose: A New Way to Store the Sperm for the Preservation of Canine Reproductive Function

**DOI:** 10.3390/ani10040653

**Published:** 2020-04-09

**Authors:** Maja Zakošek Pipan, Margret L. Casal, Nataša Šterbenc, Irma Virant Klun, Janko Mrkun

**Affiliations:** 1Clinic for Reproduction and Large Animals, Veterinary Faculty, University of Ljubljana, Gerbičeva 60, 1000 Ljubljana, Slovenia; natasa.sterbenc@vf.uni-lj.si (N.Š.); janko.mrkun@vf.uni-lj.si (J.M.); 2School of Veterinary Medicine, Clinical Sciences & Advanced Medicine, University of Pennsylvania, Philadelphia, PA 19143, USA; casalml@vet.upenn.edu; 3Division of Gynaecology and Obstetrics, University Medical Centre Ljubljana, Zaloška cesta 2, 1000 Ljubljana, Slovenia; irma.virant@kclj.si

**Keywords:** Vitrification, soy lecithin, sucrose, semen preservation, canine

## Abstract

**Simple Summary:**

Soy lecithin and sucrose were used at different concentrations to develop and compare different vitrification methods for the cryopreservation of canine semen. All of the sperm quality characteristics were effectively preserved after devitrification when vitrification extenders containing soy lecithin at 1% and 0.25 M sucrose were used. The results suggest that vitrification is effective, fast, and simple for cryopreservation of canine semen. Furthermore, the use of soy lecithin in lieu of animal proteins (e.g., egg yolk) facilitates semen shipping to countries with strict import requirements.

**Abstract:**

A challenge in freezing semen for short and long-term availability is avoiding damage to intact spermatozoa caused by the freezing process. Vitrification protocols provide better results through less manipulation of semen and shorter freezing time compared to slow freezing techniques. Our research was aimed at improving vitrification methods for canine semen. Semen quality was determined in 20 ejaculates after collection. Each ejaculate was divided into eight aliquots, each with a different extender. The control extender contained TRIS, citric acid, fructose, and antibiotics. Soy lecithin and sucrose were added to the control extender at different concentrations to make up the test extenders and final concentration of 50 × 10^6^ spermatozoa/mL. From each group, a 33 µL (1.65 × 10^6^ spermatozoa) suspension of spermatozoa was dropped directly into liquid nitrogen and devitrified at least one week later and evaluated as before. Soy lecithin at 1% and 0.25 M sucrose added to the base vitrification media effectively preserved all sperm qualities. Our results demonstrate the effectiveness of our methods. Vitrification media containing sucrose and soy lecithin cause a minimal decline in quality of canine semen after devitrification. Furthermore, extenders used in our research did not contain egg yolk, which was replaced by soy lecithin, thus allowing for ease of shipping to other countries with strict requirements.

## 1. Introduction

For decades, the semen freezing process in humans and animals has not changed much, and programmable slow-cooling methods are still commonly being used [1,2,3]. These techniques require specialized equipment and media, are time consuming, and often lead to spermatozoa damage, mostly because of ice crystallization, but also due to osmotic and chilling injury [4].

One way of avoiding damage caused by crystal formation is semen vitrification [5], which has been studied in different animal species and in humans as an alternative to conventional freezing [1,5,6,7,8,9,10]. Vitrification has been used widely for embryo storage [3] and has been a scope of research in the last decade for sperm cryopreservation [11]. This method allows for the extremely rapid freezing of samples and was originally developed in the early 1900s using semen from birds. Despite these efforts, sperm vitrification has not gained popularity so far, and encouraging results have not yet been established in most species studied [5]. Effective vitrification methods have only been developed in felids, and the successful nature of this procedure is evidenced by litters having been produced in captive exotic and domestic cats [6,12].

In dogs, few studies using vitrification protocol have been published [7,8,13,14], and conventional freezing methods are still primarily used for cryopreservation of canine semen. One weakness of vitrification methods described so far is that high concentrations of cryoprotective agents (CPAs) are being used in order to achieve successful cryopreservation. However, sperm cells are very sensitive to CPA [15]. Isachenko et al. [2] described a new method of human sperm vitrification without the use of conventional permeable CPA. Different combinations of carbohydrates (sucrose and trehalose) and proteins (human serum albumin) were used instead, and their results showed that slow freezing and vitrification were both similar in regard to sperm motility and DNA integrity [2]. While the study from Kim et al. [7] did not reveal acceptable motility and viability results when canine semen in egg-yolk-based extenders was directly plunged into liquid nitrogen. Nouri Gharajelar et al. [13] reported acceptable viability and motility with their vitrification process based on milk or egg yolk extenders. Sanchez et al. [14] used sucrose and trehalose in the vitrification media for canine semen and demonstrated that the addition of 250 mM sucrose yielded the best sperm quality results—better preservation of viability and acrosome integrity—yet the motility was still not acceptable. In all of these studies, a foreign animal protein (bovine serum albumin, milk, or egg yolk) was used in test extenders, which can present a problem with international shipment of frozen semen. In a recent study on canine semen, soy lecithin was found to be a viable alternative to egg yolk with a similar effect on mitochondrial activity and motility of cryopreserved dog semen. Moreover, samples frozen in soy-lecithin-based extenders resulted in lower lipid peroxidation compared to egg yolk extenders, which is important since oxidative stress is one of the major problems in stored semen [16]. 

The objective of this study was to improve current vitrification methods for canine semen using sucrose added to the basic canine semen extender and replacing foreign animal proteins with soy lecithin.

## 2. Materials and Methods 

### 2.1. Semen Collection

Twenty ejaculates from 19 dogs from a research colony at the University of Pennsylvania were used (IACUC protocol number 806310). Dogs of distinct breeds and body weights were obtained and evaluated. Dogs were collected by the same operator (to eliminate any variation in semen quality resulting from collection technique). One dog was collected twice, and between first and second collection, there was a three-week resting period. The collection was performed in the presence of a bitch in heat to provide stimulation. The semen was collected by digital stimulation using an artificial vagina, with a sterile pre-warmed (37 °C) glass tube attached to the end.

### 2.2. Basic Semen Evaluation

Standard semen evaluation was performed immediately after collection and before processing to ensure that the samples were normal (volume, motility, progressive motility, concentration, morphology). Only semen samples that fulfilled the requirements for normal, healthy, and fertile stud dogs were used in the study: ≥75% estimated progressive motility, ≥65% normal morphology, and ≥ 250 × 10⁶ total spermatozoa per collection [17,18].

Motility and progressive motility of spermatozoa were evaluated by computer assisted sperm analysis (Hamilton Thorne IVOS 10.2; Hamilton Thorne Research, Beverly, MA, USA) with a Makler counting chamber (Sefi Medical Instruments, Haifa, Israel). Parameters for the computer assisted system analyzer (CASA) were adjusted to accommodate canine semen samples according to protocols already in place [14]. Briefly, 10 µL of each semen sample was loaded onto the Makler chambered slide and placed into the CASA. At least six fields were counted. Each sample was assessed in duplicates. Sperm motion parameters shown by CASA were motility (M), progressive motility (PM), curvilinear velocity (CLV), mean path velocity (APV), progressive speed (SLV), and lateral head displacement (LHD).

Concentrations were measured using a spectrophotometer (SpermaCue®, Minitüb, Tiefenbach, Germany) in duplicates according to the manufacturer’s instructions. Both CASA systems and SpermaCue® have been described for assessment of concentration of spermatozoa in dogs. However, the gold standard remains a hemocytometer, which has been reported to be more accurate than CASA systems [17]. In our preliminary study we compared canine semen concentrations measured with Neubauer counting chamber, CASA system, and SpermaCue®. Each ejaculate was assessed five times with all three methods. Repeatability, expressed by coefficient of variation (CV), was determined for each method. SpermaCue® yielded the lowest CV (3.8%) (Appendix A.). Even though, repeatability was consistent and reliable also with CASA system, CASA was overestimating the concentration of spermatozoa. It was shown previously that Makler chamber in CASA system can significantly overestimate sperm concentration. One of the possible causes reported was that CASA sometimes could not separate sperm from the bright grids in a Makler counting chamber [19].

Morphology of 200 spermatozoa was assessed in semen samples after Giemsa staining in duplicates. A 5 μL aliquot of fresh canine semen was placed on the slide, smeared, and allowed to air-dry. Staining was performed according to Watson et al. [20] and rinsed with distilled water. Slides with fixed, and stained semen samples were examined by optical microscopy at 1000× magnification. Total morphological abnormal spermatozoa were divided into head abnormalities, acrosome abnormalities, neck/midpiece abnormalities, bent and coiled tails, and incidence of proximal, and medial to distal cytoplasmic droplets. 

Viability of semen samples was assessed by mixing 5 μL of fresh canine semen with 5 μL of eosin-nigrosin (Morphology stain, Society of Theriogenology, Pike Road, AL, US) and allowed to air dry. At least 200 spermatozoa were counted in duplicates under direct bright light and oil immersion at magnification of ×1000.

### 2.3. Other Semen Analysis

#### 2.3.1. Acrosome Assessment

The sperm acrosomes were evaluated using the Spermac™ Sperm Staining Kit (Minitüb, Tiefenbach, Germany)), which allows for clear differentiation between the post-acrosomal region and the nuclear portion of the head, the acrosome, and the midpiece of spermatozoa. At least 200 spermatozoa were counted in duplicates under direct bright light and oil immersion at magnification of ×100.

#### 2.3.2. Hypoosmotic Swelling Test (HOST)

The hypoosmotic swelling test (HOST) assay was applied to evaluate the sperm tail membrane integrity as an indirect measure of sperm functionality. It was performed by gently mixing 10μL of semen sample with 90μL of hypoosmotic solution (150 mOsm/kg sodium citrate × 2H_2_O and fructose) both warmed at 37 °C. After one hour of incubation at 37 °C in water bath, 200 spermatozoa were examined under light microscope at magnification of ×400. Spermatozoa were considered HOST positive if they showed signs of swelling, as described by Hishinuma and Sekine [21] (Figure 1).

#### 2.3.3. DNA Fragmentation

Sperm DNA integrity was evaluated using a commercial test (Sperm Canis-Halomax; Halotech DNA SL, Madrid, Spain) based on the Sperm Dispersion test and specifically designed for canine spermatozoa. The sperm samples were processed according to the manufacturer instructions. The slides were then stained with a commercial fluorescence microscopy green staining kit (Halotech DNA, Madrid, Spain) according to the instructions. Sperm chromatin dispersion was evaluated using fluorescent filter (Olympus U-MNIBA3; excitation: 497 nm and emission: 520 nm) and 400× magnification (Olympus BX40, Olympus, Hamburg, Germany). A minimum of 300 spermatozoa were counted per semen sample. Samples were examined in duplicates. 

The spermatozoa were classified into three categories according to the shape of the halo effect: (1) normal halo, spermatozoa showing clearly visible halo around the head similar to the diameter of the head—classified as spermatozoa with normal DNA; (2) small halo or absent halo, spermatozoa showing a small halo spotted around the head or complete absence of halo; and (3) large scattered halo, spermatozoa showing a very large and scattered halo around the head. Spermatozoa from groups (2) and (3) were classified together as spermatozoa with damaged DNA.

### 2.4. Experimental Setup

Semen was collected from dogs and evaluated as described above. The semen was then centrifuged at 700× *g* for 10 min at room temperature. The supernatant was discarded, and the semen pellet was resuspended in the control extender to a concentration of 50 × 10^6^/mL. Each ejaculate was then aliquoted into eight vials. The samples were again centrifuged at 700× *g* for 10 min at room temperature, the supernatant discarded, and the extenders described in Table 1 were added to the pellet to a concentration of 50 × 10^6^/mL. The samples were then allowed to equilibrate for 15 min at room temperature.

### 2.5. Preparation of Extenders Used in the Study

All chemicals and media were obtained from Sigma-Aldrich (St. Louis, MO, USA) unless mentioned otherwise.

Ejaculates were divided into 16 aliquots, and different vitrification media labeled A–G were added (Table 1). The control extender consisted of TRIS, citric acid, fructose, and antibiotics, as can be seen in Table 1.

According to the Dalmazzo et al. [16], who was comparing different forms of soy lecithin on frozen canine semen samples, Solec F (Solae F, Solae Company, St. Louis, MO, USA) exhibited best results and was therefore used in our study. In our preliminary study, the best concentration of soy lecithin to be added to the vitrification media was determined and the following concentrations: 1%, 2%, 3% and 4% (wt/vol) were compared on five canine semen samples. The parameters evaluated were viability, percentage of progressive motile spermatozoa, and morphology before and after devitrification. It was found out that 1% soy lecithin concentrations were giving the best results (Casal ML., unpublished data) (Appendix A).

Soy lecithin (Solae F, Solae Company, St. Louis, MO, USA) and sucrose were added to the control extender at different concentration to make up the test extenders. For Extender A, only soy lecithin (1%) was added. Extenders B, C, and D contained different concentrations of sucrose (0.2 M, 0.25 M, and 0.3 M, respectively) but no soy lecithin. Extenders E, F, and G contained soy lecithin (1%) and different concentrations of sucrose (0.2 M, 0.25 M, and 0.3 M, respectively). Preparation of extenders is shown in the Table 1. Double distilled water was added to make 100 mL of each extender, and the pH and osmolality were adjusted to 6.5 and 385 mOsm/kg. After preparation of each extender, the extenders were centrifuged at 2900× *g* for 15 min at room temperature to remove all solids. Thereafter, the extenders were filtered successively through a 0.8 µm and 0.4 µm filter for sterility. 

### 2.6. Vitrification Process

Vitrification process was carried out using slightly modified protocol previously described by Isachenko et al. [1] and Sanchez et al. [8]. After addition of vitrification media, samples were maintained at 5 °C for 30 min (equilibration time). The diluted samples were pipetted and plunged into liquid N_2_ (LN_2_) at a volume of 33 µL (spermatozoa concentration in each drop was 1.65 × 10^6^) from a height of 11 cm. Care was taken to drop one in at a time, in order to avoid the droplets sticking together. The solidified samples were stored in cryo-vials (2 mL, CM LAB, Montréal, Canada) in LN_2_ for at least a week until devitrification for evaluation.

### 2.7. Devitrification Procedure

For devitrification procedure, five solidified sperm pellets were added to the 1 mL of control medium containing 1% bovine serum albumin (BSA) (Sigma-Aldrich, Merck, Darmstadt, Germany) that has been pre-warmed to 37 °C. Sperm pellets with medium were then incubated at that temperature for 2 min. After warming, the sperm parameters were evaluated as described above. After two minutes of incubation, motility and progressive motility were assessed with the CASA system. Slides for viability and morphology were prepared and the semen was immediately prepared for HOST incubation and DNA fragmentation test. All of the analysis started within 10 min after devitrification.

### 2.8. Statistics

#### 2.8.1. Sample Size

Sample size was calculated before the beginning of the study using a formula for group comparison and repeated measurements [22]. Since the statistical characteristic that has been measured was numerical, the mean value for the population that was based on published literature was estimated. Most of the studies for semen evaluation in animals are using 15–20 ejaculates in order to obtain a statistically significant result.

Our study aimed to compare semen quality using eight different extenders with two measurements (before and after vitrification). Two comparisons were made—comparing the quality of fresh semen samples in different extenders to that after devitrification and comparing different semen extenders with each other. The following formula was used to calculate the smallest sample size. Considering the error DF (degrees of freedom), the sample sizes per group were calculated as follows: Minimum n = 10/(2 (two measurements) × 8 (different extenders)) + 1 = 1.63 = rounded up to 2 samples/group.(1)
Maximum n = 20/(2 × 8) + 1 = 2.25 = rounded down to 2 animals/group resulting in equal sample sizes for the minimum and maximum.(2)
N: Minimum (Maximum) = Minimum (Maximum) n × 8 = 2 × 8 = 16 samples(3)

According to this calculation, at least 16 samples were needed to give us statistically significant results.

#### 2.8.2. Statistical Analysis

Data are presented as mean ± standard deviation (SD) and were compared using a Multilevel Mixed-Effect linear regression assay and Mann Whitney U test (GraphPad Software, San Diego, CA, USA). Normal distribution of data was tested by Kolmogorov-Smirnov-Test. The results obtained for different semen extenders were normally distributed and compared using One Way Repeated Measures Analysis of Variance. In case of a significant difference between groups of samples, values were compared by the Tukey test. Differences between semen extenders were compared using a paired Student’s t-test in case of normal distribution. Wilcoxon test was used in the case that data were not normally distributed, which was only the case when fresh samples were compared to vitrification groups. Statistically, significant difference was set at *p* < 0.05.

## 3. Results

Initial sperm quality variables from the ejaculates used in this study had the following mean values: sperm concentration 215.73 ± 66.54 million sperm/mL (range: 177.5–349.5 million sperm/mL), total motility 86.50 ± 3.28% (range: 80.0–90.0%), progressive motility 77.50 ± 3.03% (range: 75.0–85.0%), normal morphology 77.39 ± 10.51% (range: 65.0–93.0%), acrosome intact sperm 94.17 ± 4.4% (range: 87.5–97%), viability 94.35 ± 3.3% (range: 88.0–98.0%) and integrity of the tail plasma membrane 92.1 ± 4.72% (range: 85.5–97.0%).

### 3.1. Semen Motility and Progressive Motility 

Fresh sperm samples showed significantly higher total and progressive motility compared to all other samples (*p* < 0.05). 

The cryoprotective effect of sucrose supplemented with soy lecithin on motility and progressive motility after vitrification are illustrated in Figure 2 and Figure 3. The motility and progressive motility after devitrification were significantly higher in the sperm vitrified with sucrose and soy lecithin compared to the control and samples containing only sucrose or only soy lecithin (*p* < 0.05). The highest semen motility and progressive motility were obtained when 0.25 M sucrose and 1 g soy lecithin were added into the vitrification medium (Extender F) and were statistically significantly better even when comparing to extender E and extender G, which contained different concentration of sucrose (*p* < 0.05). 

Motility of fresh semen samples compared to all other extenders after devitrification and marked by different uppercase letters, differed significantly (*p* < 0.05). Motility of semen quality after devitrification in different extenders, marked by different lowercase letters, differed significantly (*p* < 0.05).

Forward progressive motility of fresh semen samples compared to all other extenders after devitrification and marked by different uppercase letters, differed significantly (*p* < 0.05). Forward progressive motility of semen quality after devitrification in different extenders, marked by different lowercase letters, differed significantly (*p* < 0.05). 

### 3.2. Viability

Statistically significant differences were observed in viability of sperm vitrified in control medium compared to all other vitrification media (Figure 4). However, the percentage of viable sperm after devitrification showed statistically significant difference and was improved by addition of soy lecithin and sucrose (0.2; 0.25M and 0.3M) to the vitrification medium compared to sperm samples without one of those two components in the medium (*p* < 0.05).

Viability of fresh semen samples compared to all other extenders after devitrification and marked by different uppercase letters, differed significantly (*p* < 0.05). Viability of semen quality after devitrification in different extenders, marked by different lowercase letters, differed significantly (*p* < 0.05).

### 3.3. Normal Morphology and Acrosome-Intact Sperm

Morphological abnormalities in spermatozoa increased during vitrification and storage and reached a significantly higher level in comparison to fresh sperm samples (*p* < 0.05) (Figure 5). Vitrification of sperm in a medium containing soy lecithin yielded a significantly lower percentage of spermatozoa with abnormal morphology compared to samples vitrified without this protein in the vitrification media (*p* < 0.05) (Table 2). However, there was a higher level of acrosome-intact spermatozoa (determined with Spermac stain) when vitrification media contained soy lecithin and sucrose (*p* < 0.05) in comparison to media without these two components (Figure 6). There were also statistically significant differences between sperm samples that were vitrified in media containing different concentrations of sucrose, whereby morphology and acrosome integrity was highest for those samples preserved in extender E (*p* < 0.05).

Morphological differences in fresh semen samples compared to all other extenders after devitrification and marked by different uppercase letters differed significantly (*p* < 0.05). Morphological differences in semen quality after devitrification in different extenders, marked by different lowercase letters, differed significantly (*p* < 0.05). 

Acrosomal abnormalities in fresh semen compared to all other extenders after devitrification and marked by different uppercase letters differed significantly (*p* < 0.05). Acrosomal abnormalities in fresh semen after devitrification in different extenders, marked by different lowercase letters, differed significantly (*p* < 0.05).

### 3.4. Tail Membrane Integrity

The percentage of spermatozoa with intact tail membrane was significantly higher in fresh sperm samples compared to all other groups of cryostored samples (*p* < 0.05). Spermatozoa that were vitrified in medium containing sucrose and soy lecithin (E, F and G) preserved their tail membrane integrity significantly better compared to other groups of frozen semen (Figure 7; *p* < 0.05). Among these three groups (E, F, and G), the tail membrane integrity was best preserved in sperm which was vitrified in extender F and was significantly better compared to extender G (*p* < 0.05). However, the difference in sperm tail integrity was not significant between extenders E and F (*p* > 0.05).

Sperm tail membrane integrity in fresh samples compared to all other extenders after devitrification and marked by different uppercase letters, differed significantly (*p* < 0.05). Sperm tail membrane integrity after devitrification in different extenders, marked by different lowercase letters, differed significantly (*p* < 0.05). 

### 3.5. DNA Integrity

Sperm DNA fragmentation after devitrification was significantly decreased by the addition of sucrose and soy lecithin to the vitrification medium (Figure 8). DNA integrity was best preserved when 0.25 M sucrose was added to the vitrification medium. Sperm samples which were vitrified in medium (extender) F did not differ significantly from fresh sperm samples in terms of sperm DNA fragmentation, whereas all other groups of cryostored semen showed significantly higher levels of sperm DNA fragmentation (*p* < 0.05) after devitrification.

Fragmented DNA in fresh semen samples compared to all other extenders after devitrification and marked by different uppercase letters differed significantly (*p* < 0.05). Fragmented DNA in semen samples after devitrification in different extenders, marked by different lowercase letters, differed significantly (*p* < 0.05). 

## 4. Discussion

The present study was aimed at improving vitrification methods for dog spermatozoa preservation. We found that soy lecithin can be used as a cryoprotective agent in the extender as an alternative to other proteins of animal origin, thereby simplifying transport of vitrified dog semen to all countries in the world. The key discovery was that sucrose and addition of soy lecithin were important components that lead to the success of canine sperm vitrification. Furthermore, in the present study, preservation of sperm quality post vitrification was better than in other canine studies using vitrification protocols [7,8,14].

Sucrose has been utilized as a nonpermeable cryoprotectant in the vitrification of sperm in several species, both in the presence and absence of animal proteins [8,23]. Sanchez et al. [8] utilized sucrose for dog spermatozoa vitrification, however, their extender medium also contained BSA, an animal protein. As previously stated, BSA and egg-yolk-based extenders that incorporate animal proteins for sperm stability are not ideal for international shipping. In our study, canine sperm quality was not preserved after devitrification, regardless of sucrose concentration when sucrose only extenders were used. In humans, sucrose alone adequately preserves semen viability and motility [24]. Sucrose alone has also been demonstrated to support successful vitrification and ultimately fertilization in wild goats [23], mouflons [10], and stallions [25]. However, our results are in agreement with other studies in which some protein source is necessary to support survival during the initial cold shock of cryopreservation. Caturla-Sanchez et al [14] reported 44% viability of canine spermatozoa vitrified in an egg yolk-based extender supplemented with 0.25M sucrose. Here, we observed 59% viability with the soy-lecithin-based extender and the same sucrose concentration. It has previously been postulated that precipitates from high egg yolk concentrations act to impede contact between sugar molecules and spermatozoa [26], which may account for the lower viability of post-thaw sperm in that study—although higher concentrations of sucrose have also resulted in lower sperm motility and viability in dogs [14] and mice [27] with egg yolk based extenders.

Egg yolk, BSA, and lecithin are proteins known for their protective qualities against the initial cold shock during the freezing process. Soy lecithin can also serve as an antioxidant and can protect semen from oxidative stress resulting in higher viability and motility after cryopreservation [16]. A recent study in dogs showed that soy lecithin at low concentrations (0.01%, 0.05%, and 0.1%) was not able to maintain sperm characteristics when compared to egg-yolk-based extenders [28]. However, increasing the concentration of soy lecithin to 1% led to promising results on post-thaw semen viability and acrosome integrity in the current study, which is in agreement with previous studies [16,29,30]. This finding supports previous suggestions that high concentrations of soy lecithin are necessary to maintain canine sperm characteristics (i.e., motility kinetics) during storage [28]. Besides that, the type of soy lecithin used as a base in the extender seems to be very important, since different lecithin products contain different concentrations of phosphatidylcholine [16], which has an important role in incorporating choline in the polar region of spermatozoa plasma membrane and plays an active role in sperm capacitation and acrosome reaction. Lipids are also essential for sperm viability [31]. The lecithin, Solec F, used in our study has been shown to preserve semen quality best when compared to other commercial products [16].

Vitrification has been successfully used in freezing of oocytes and embryos with the benefit of being a quick process that does not require expensive equipment [8]. Furthermore, by directly plunging samples into liquid nitrogen, solidification of living cells is achieved without intracellular ice crystallization [32]. However, spermatozoa are osmotically fragile cells and cannot withstand the lethal effect of permeable cryoprotectants at high concentrations needed for successful vitrification [22]. Therefore, it was proposed, that far lower concentrations of permeable CPAs or nonpermeable CPAs may play a leading role in sperm vitrification [11]. To achieve this, droplets of semen in vitrification media were directly plunged into the liquid nitrogen in our study. This procedure was first described by Isachenko et al. [23] for human semen and soon after also for canine sperm vitrification [8]. Recently, studies using human sperm vitrification protocols showed superior results in motility, viability, DNA structure, acrosome and morphology when compared to conventional freezing protocols [11,32,33]. In a study by Le et al. [34], morphology in spermatozoa was better preserved after devitrification with less defects observed when compared to conventional freezing. In our study, morphological changes in spermatozoa increased during vitrification and storage at the cost of intact acrosomes and sperm tails. Osmotic changes during the cryopreservation are associated with coiled tails. However, mechanical changes are considered the main cause of cell damage when vitrification is used [35]. In the present study, the percentage of intact acrosomes, which usually decreases rapidly in the canine species once spermatozoa are separated from the seminal plasma [36], remained high in all vitrified samples. The same was observed in a previous study where vitrification of canine spermatozoa resulted in a significantly higher level of acrosome intact spermatozoa compared to conventional freezing, suggesting that the physio-chemistry of vitrification in the presence of disaccharides is beneficial for acrosome integrity [14].

In the present study, neither soy lecithin (extender A) or sucrose (extenders B, C, and D) alone was sufficient to adequately preserve sperm quality after devitrification. Our findings are supported by other recent studies in which the addition of protein to sucrose was needed to adequately preserve motility and viability of spermatozoa in dogs and humans post vitrification [8,23]. In our study, vitrification media containing soy lecithin and various concentrations of sucrose resulted in a significantly better semen quality after devitrification. Vitrification media with soy lecithin and 0.25 M sucrose had the highest proportion of total motile, progressively motile, morphologically normal and viable spermatozoa. There was also a higher proportion of spermatozoa with intact DNA and normal plasma membrane integrity. No significant difference in DNA fragmentation was observed with this media when compared to fresh semen, even though DNA fragmentation significantly increased in all other groups of cryostored semen. This finding is in agreement with a study in dogs where >95% DNA integrity preservation was obtained after vitrification of spermatozoa [8]. Quality of spermatozoal DNA is very important for fertilization and the transmission of undamaged genetic material to the next generation. Vitrification in our study was achieved through direct plunging of semen into the liquid nitrogen without the presence of permeable cryoprotectant, which is deemed to be responsible for DNA damage. Previous studies reported that this method of vitrification enabled adequate preservation of DNA integrity [2,8,13]. 

Our study demonstrated that vitrification of canine spermatozoa in an extender containing soy lecithin and sucrose adequately preserved important sperm physiological attributes, including viability, motility, progressive motility, morphology, acrosome, sperm tail membrane integrity, and DNA integrity. These results indicate that vitrification of sperm can be an effective and efficient method of canine semen cryopreservation. This method is simple, fast, and does not require sophisticated equipment. Soy lecithin is an attractive protein to replace animal proteins such as albumin and egg yolk in vitrification media to achieve good quality of sperm after devitrification and to safely transport the sperm samples worldwide. Since our primary goal was to compare semen cryosurvival using extenders with soy lecithin and different concentrations of sucrose, the true measure of successful vitrification (i.e., fertility study) was not performed. Future studies comparing the method of vitrification described here with conventional semen freezing methods are needed to fully evaluate its efficacy.

## Figures and Tables

**Figure 1 animals-10-00653-f001:**
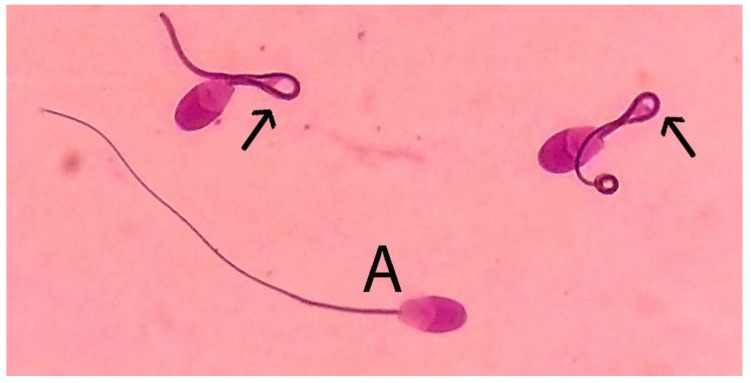
Spermatozoa after hypoosmotic swelling test. Arrow: hypoosmotic swelling test (HOST) positive spermatozoa that show signs of tail swelling; A: HOST negative spermatozoa without swelling.

**Figure 2 animals-10-00653-f002:**
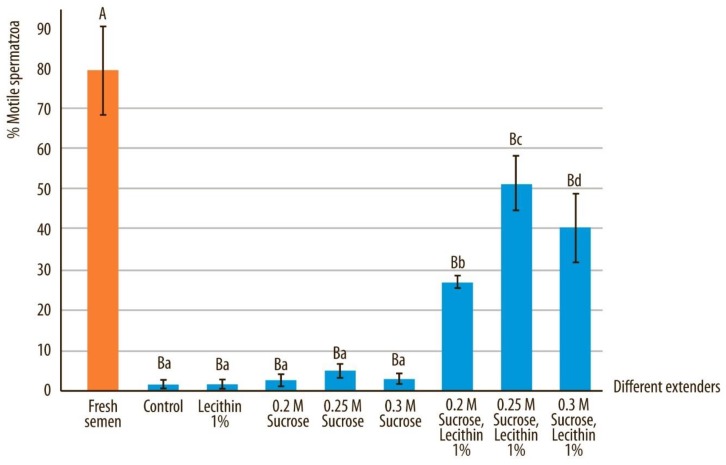
Total motility in canine sperm after devitrification regarding the vitrification media. Data are expressed as mean ± SD.

**Figure 3 animals-10-00653-f003:**
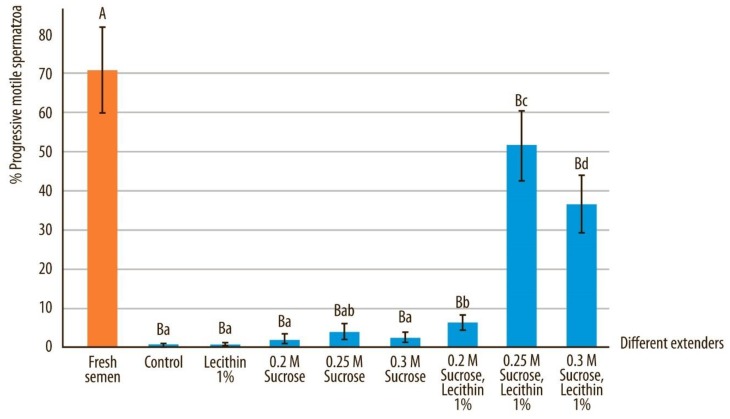
Progressive motility in canine sperm after devitrification regarding different vitrification media. Data are expressed as mean ± SD.

**Figure 4 animals-10-00653-f004:**
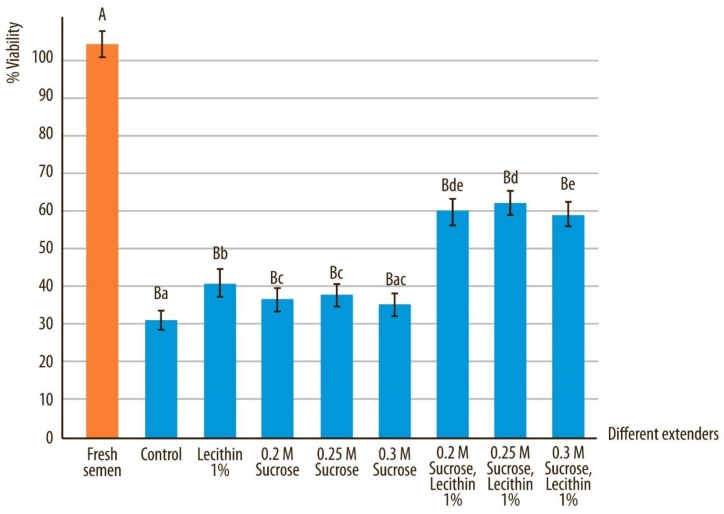
Viability in canine sperm after devitrification regarding different vitrification media. Data are expressed as mean ± SD.

**Figure 5 animals-10-00653-f005:**
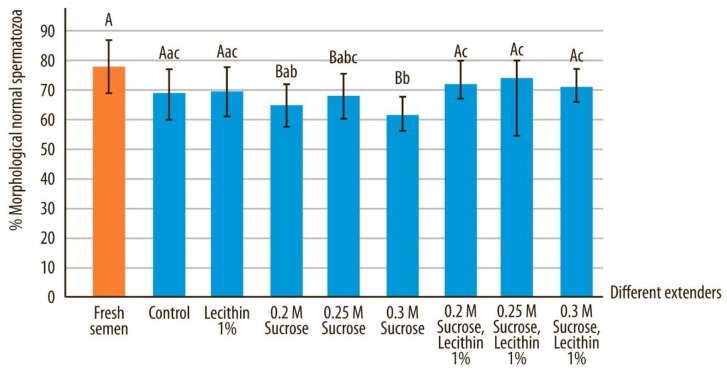
Normal morphology in canine sperm after devitrification regarding different vitrification media. Data are expressed as mean ± SD.

**Figure 6 animals-10-00653-f006:**
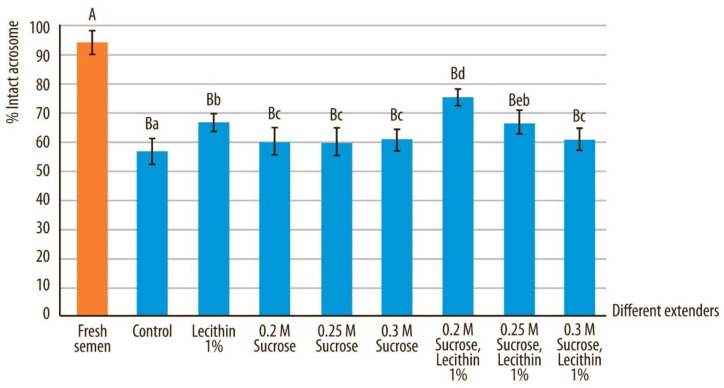
Intact acrosome based on Spermac staining in canine sperm after devitrification regarding different vitrification media. Data are expressed as mean ± SD.

**Figure 7 animals-10-00653-f007:**
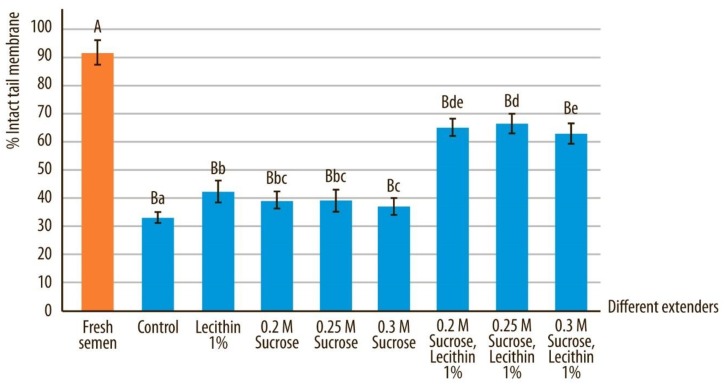
Sperm tail membrane integrity in canine spermatozoa after devitrification regarding different vitrification media. Data are expressed as mean ± SD.

**Figure 8 animals-10-00653-f008:**
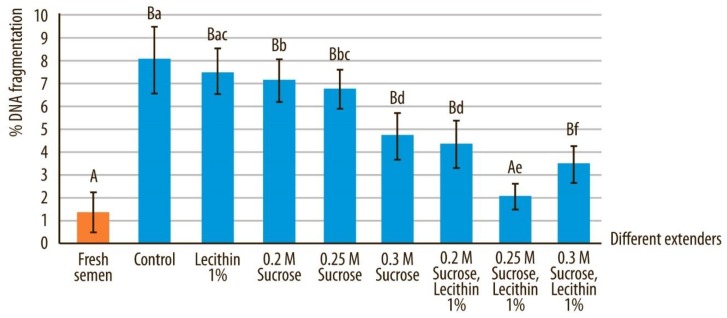
Percentage of spermatozoa with fragmented DNA in canine sperm after devitrification regarding different vitrification media. Data are expressed as mean ± SD.

**Table 1 animals-10-00653-t001:** Preparation of extenders used in the study.

	Control	Ext. A	Ext. B	Ext. C	Ext. D	Ext. E	Ext. F	Ext. G
**Tris**	2.4 g	2.4 g	2.4 g	2.4 g	2.4 g	2.4 g	2.4 g	2.4 g
**Citric Acid**	1.4 g	1.4 g	1.4 g	1.4 g	1.4 g	1.4 g	1.4 g	1.4 g
**Fructose**	1.0 g	1.0 g	1.0 g	1.0 g	1.0 g	1.0 g	1.0 g	1.0 g
**Streptomycin**	5 µg	5 µg	5 µg	5 µg	5 µg	5 µg	5 µg	5 µg
**Penicillin**	10kIU	10kIU	10kIU	10kIU	10kIU	10kIU	10kIU	10kIU
**Sucrose**	None	None	0.2 M	0.25 M	0.3 M	0.2 M	0.25 M	0.3 M
**Soy lecithin**	None	1%	None	None	None	1%	1%	1%

Ext.—extender, unit M is for molarity, unit kIU is for ‘kilo’ *international unit.*

**Table 2 animals-10-00653-t002:** Different types of morphological abnormalities in different extenders expressed as mean ± SD.

Morphological Defects	Fresh Semen	Control	Ext. A	Ext. B	Ext. C	Ext. D	Ext. E	Ext. F	Ext. G
**Head**	2.11 ± 2.71^A^	2.75 ± 2.12 ^Aa^	4.85 ± 1.30 ^Bb^	3.30 ± 2.75 ^Aa^	2.93 ± 2.35 ^Aa^	3.95 ± 3.05 ^Bb^	2.33 ± 2.03 ^Aa^	2.20 ± 1.96 ^Aa^	2.45 ± 1.87 ^Aa^
**Detached acrosome**	0.78 ± 1.23 ^A^	4.85 ± 1.30 ^Ba^	4.75 ±1.49 ^Ba^	5.38 ± 1.30 ^Bb^	5.20 ± 1.06 ^Bab^	6.25 ± 1.40 ^Bc^	4.73 ± 1.26 ^Ba^	4.43 ± 0.85 ^Ba^	5.03 ± 1.28 ^Ba,b^
**Midpiece defects**	2.93 ± 2.91 ^A^	2.83 ± 1.89 ^Aa^	3.00 ± 2.99 ^Aa^	3.25 ± 2.56 ^Aa^	3.13 ± 2.34 ^Aa^	3.33 ± 2.68 ^Aa^	2.58 ± 2.68 ^Aa^	2.38 ± 2.74 ^Aa^	3.03 ± 3.28 ^Aa^
**Proximal droplets**	2.33 ±1.75 ^A^	4.03 ± 2.38 ^Aa^	3.43 ± 2.23 ^Aa^	4.03 ± 2.00 ^Aa^	3.83 ± 1.87 ^Aa^	4.63 ± 2.16 ^a^	3.05 ± 2.17 ^Aa^	3.13 ± 2.34 ^Aa^	3.10 ± 2.15 ^Aa^
**Distal droplets**	4.00 ± 5.60^A^	5.73 ± 3.81 ^Aa^	4.95 ± 3.48 ^Ab^	5.43 ± 3.18 ^Aa^	5.28 ± 2.61 ^Aa^	6.10 ± 2.91 ^Aa^	4.30 ± 3.35 ^Ab^	4.10 ± 2.90 ^Ab^	4.35 ± 3.52 ^Ab^
**Bent tail**	2.78 ± 1.79 ^A^	5.20 ± 2.73 ^Ba^	5.45 ± 2.76 ^Ba^	7.03 ± 3.30 ^Bb^	5.60 ± 2.90 ^Ba^	7.70 ± 3.51 ^Bb^	5.15 ± 2.68 ^Ba^	4.70 ± 2.78 ^Ba^	5.95 ± 3.04 ^Ba^
**Coiled tail**	6.06 ± 6.71 ^A^	6.95 ± 4.41 ^Aa^	6.80 ± 4.75 ^Aa^	7.63 ± 3.98 ^Aa^	6.95 ± 4.29 ^Aa^	7.93 ± 3.79 ^Aa^	6.28 ± 3.94 ^Aa^	5.68 ± 3.43 ^Aa^	6.13 ± 3.40 ^Aa^
**Normal sperm**	77.39 ± 10.51 ^A^	68.85 ± 8.01 ^Bac^	69.73 ± 8.38 ^Bac^	64.7 0 ± 7.22 ^Bab^	68.20 ± 7.58 ^Babc^	61.40 ± 6.68 ^Bb^	72.18 ± 7.51 ^Ac^	74.15 ± 6.02 ^Ac^	70.48 ± 6.92 ^Bc^

Ext.—extender. Uppercase letters (A, B): Morphological differences in fresh semen samples compared to all other extenders after devitrification and marked by different uppercase letters, differed significantly (*p* < 0.05). Lowercase letters (a, b, c): Morphological differences in semen quality after devitrification in different extenders, marked by different lowercase letters, differed significantly (*p* < 0.05).

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
