# Peer review of "Vitrification Using Soy Lecithin and Sucrose: A New Way to Store the Sperm for the Preservation of Canine Reproductive Function"

_animals, 2020, doi:10.3390/ani10040653_

Round 1
Reviewer 1 Report
In the present study, authors aim to improve current vitrification methods for canine semen using sucrose added to the basic canine semen extender and to suggest to replace foreign animal proteins with soy lecithin for an easier shipment of semen to foreign countries with stric requirements. Formally the manuscript appears properly prepared. The aim is clear, and the interest is current in canine reproduction. However the study design presents an important trouble. To properly support the authors' final thesis a control group would have been necessary in the study design allowing to compare the obtained results to a validated conventional semen freezing protocol.
Specific comments:
Simple summary: please shorten and rephrase the simple summary that according to the author guidelines should consists of a short single paragraph written for a lay audience focusing on the aims and main results of the research to elucidate how the conclusions from the study will be valuable to the scientific society. Please do not duplicate background from the abstract.
line 38: please elucidate from the abstract the sperm concentration you worked with and you used in the 33ul suspension of spermatozoa dropped in liquid nitrogen;
line 101: please provide Ref for normal range considered in your paper;
line 115: please support the decision to measure concentration by the SpermaCue instead of considering results obtained by the CASA System you used;
Section 2.8: please consider to better clarify the statistical analysis: readers should know if your results were normally distributed or not, and the statistical test for each result reported in text and graphs should be clearly stated;
Section 3.3: data regarding different types of morphological abnormalities from the different extenders could be interesting and should be reported to the readers and the effect of the different extenders might be deeper analyzed in discussion;
line 362: As authors report, effectively the number of semen samples evaluated is small. Please provide statistical support for the small number of samples recruited in order to demonstrate the strengh of the obtained results.
line 363: As reported above this concerns the main trouble of the research and should be more emphatised as an objective important limit of the study that weakens the strengh of the conclusions.
Reviewer 2 Report
The manuscript “Vitrification using soy lecithin and sucrose: a new way to store the sperm for the preservation of canine reproductive function” represents important work in the effort to develop adequate egg-yolk alternatives for gamete cryopreservation. The use of multiple metrics, including sperm motility, DNA integrity, and acrosome and tail membrane integrity post-thaw canine semen following vitrification in various concentrations of sucrose in the presence of soy lecithin, is important to the critical evaluation of this new technique. While the results themselves are compelling, the discussion needs significant improvement to orient the current study’s data with existing knowledge in dog sperm vitrification. Further, additional details are needed in the methods.
Introduction
- As a general comment, the focus on slow freezing versus vitrification in the introduction throws off the introduction, because you did not compare the two methods in the manuscript.
- Line 65 – clarify that the drawbacks described are more an issue with slow freezing
- Lines 76 – 88 detail the results of these studies in dogs more? Especially the latter, if it is the basis of the “current vitrification methods for canine semen” used in this study
Methods
- More details about the preparation of sperm slides – fixation, etc
- Line 118-119. There should be other citations available for this stain for sperm
- Lines 163-166: was this a preliminary study? How many dogs? Where are these data?
- Soy lecithin is incredibly difficult to get into solution – how can you be sure that the w/v amounts you started with are the same and/or that there was no variability between the Extender groups?
- Line 180: cite the source of this vitrification method again
- Line 187: Isnt the presence of BSA in the devitrification solution counter to the lack of animal protein goals?
Results
- Graph axes need to be labeled – also would be better to make the x axis with the amounts of sucrose and soy lecithin rather than naming the extenders (which goes for the entire manuscript)
- Id like to see the fresh (non-cryopreserved) control on the graphs as well
- Recommend an a-b-c-d system for significant differences on the graphs – the * are hard to read
- All these analyses were made within how many minutes of devitrification?
- Maybe include some representative images of the sperm tail membrane integrity assay?
Discussion
- Line 297 – what about the citation [13] paper you described in the introduction??
- Lines 307 – 3019: again, focus less on vitrification itself, since you didn’t compare vitrification to slow freezing in this study. The only part that is relevant is how your optimized vitrification compares with the best published slow freezing methods in dogs
- Has anyone else used this dropwise vitrification method for canine sperm before?
- Lines 362 – 366: I don’t think you need to point out the study faults in this way. Lack of sample volume is not a good rationale for not including desired controls
- Citation on lines 467 – where is this discussed? Line 349? It’s not numbered. There are few studies on soy lecithin in dog sperm, they should all be compared in detail against your work in the discussion. What metrics of evaluation did the other studies use. Did your sucrose+soy lecithin combination come close to the egg yolk results of other studies?
- Should also mention the Dalmazzo 2018 (https://doi.org/10.1017/S0967199418000576) paper (although chilling, it is still soy lecithin and dog sperm)
Round 2
Reviewer 1 Report
To the Authors of the submitted manuscript entitled “Vitrification using soy lecithin and sucrose: a new way to store the sperm for the preservation of canine reproductive function ”I suggested to require some further revisions in order to make the manuscript acceptable for publication in Animals.
Mainly I would like the authors to go more in depth on the following aspects:
-
Please detail results on your preliminary studies, as they represent effectively part of the M&M section of the present research as they provide the basis for your subsequent work (evaluate the possibility to include this part of the research in a different sub-chapter section of the present paper):
- concerning the best concentration of soy lecithin to be added to the vitrification media (Lines 170-177)
- concerning th assessment of the concentration of spermatozoa by Neubauer counting chamber, CASA system, and SpermaCue® (Lines 103-111);
- the statistical analysis is now clearer in M&M section, please better clarify to the readers if your results were normally distributed or not, thus I suggest to clearly state in results section otherwise to indicate the statistical test for each result reported in text and/or graphs;
- please include in the manuscript your statistical support for the small number of samples recruited (I suggest to explain clearly) in order to demonstrate the strength of expected obtained results;
Best regards
Author Response
Thank you for your thoughtful suggestions. We tried to incorporate them into the manuscript and we agree that they brought an additional value to the manuscript.
Sincerely,
Maja Z. Pipan

Reviewer 2 Report
Manuscript very much improved! The discussion section does not read smoothly though, jumps back and forth a lot and is hard to follow. Recommend reorganization prior to publication. I'm including an example outline and paragraph below, though it doesnt have to be this organization specifically, but something more structured than current status.
How you should organize your discussion
- Summary – goals, main findings of study
- Freezing overall, damages and changes between fresh and cryo in this study
- Lines 446 – 455 examples
- Sucrose only results (example option below)
- How sucrose works. For example, it didn’t seem to impair membranes/acrosomes/morphology as much in your study, but did impact motility and DNA status more?
- Why do you think it doesn’t work on its own in dogs
- Comparison with other canine research where relevant
- Lecithin only results
- How lecithin functions compared with egg yolk in other studies
- What areas it supported best in freezing in your study (acrosome? Survival? Motility? Describe your results)
- Comparison with other research where relevant
- Combination of sucrose and lecithin
- Conclusion (lines 456 – 469)
Sucrose has been utilized as a nonpermeable cryoprotectant in the vitrification of sperm in several species, both in the presence and absence of animal proteins. Sanchez et al utilized sucrose for dog spermatozoa vitrification; however, the extender medium also contained BSA, an animal protein. As previously stated, BSA and egg yolk-based extenders relying on animal proteins are not ideal for international shipping goals. In the absence of protein (i.e. sucrose only treatments), canine sperm quality was not preserved after devitrification, regardless of sucrose concentration. In humans, sucrose alone is preserves semen viability and motility [27]. Sucrose alone has also been demonstrated to support _you say “successful semen vitrification - - does this mean live births? That is ‘success’ in my definition__ in wild goats [23], mouflons [10], and stallions [26]. However, our results are in agreement with other studies in which some protein source is necessary to support survival during vitrification. __WHY – WHAT DOES PROTEIN DO DURING VIT?___ For example, Caturla-Sanchez et al [14] reported 44% viability of canine spermatozoa vitrified in an egg yolk-based extender supplemented with 0.25M sucrose. Here, we observed 59% viability with the soy lecithin based extender and the same sucrose concentration. It has previously been postulated that precipitates from high egg yolk concentrations act to impede contact between sugar molecules and spermatozoa [28], which may account for the lower viability of post-thaw sperm in that study – although higher concentrations of sucrose have also resulted in lower sperm motility and viability in dogs [14] and mice [25] with __??__ based extenders.
Similarly, here we demonstrate that soy lecithin alone, in the absence of sucrose, is also not sufficient to adequately preserve sperm quality after devitrification. In dogs, the use of soy lecithin …
Author Response
Thank you very much for your helpful suggestions. We have reorganized and rewritten the discussion section accordingly. Please look at the text below.
Sincerely,
Maja Z. Pipan
